# Mechanical Properties and Nanomotion of BT-20 and ZR-75 Breast Cancer Cells Studied by Atomic Force Microscopy and Optical Nanomotion Detection Method

**DOI:** 10.3390/cells12192362

**Published:** 2023-09-26

**Authors:** Maria N. Starodubtseva, Nastassia M. Shkliarava, Irina A. Chelnokova, María I. Villalba, Andrei Yu. Krylov, Eldar A. Nadyrov, Sandor Kasas

**Affiliations:** 1Department of Medical and Biological Physics, Gomel State Medical University, 246000 Gomel, Belarus; 2Laboratory of the Stability of Biological Systems, Radiobiology Institute of NAS of Belarus, 246007 Gomel, Belarus; anshkliarava@gmail.com (N.M.S.); irenachelnokova@gmail.com (I.A.C.); 3Laboratory of Biological Electron Microscopy, Ecole Polytechnique Fédérale de Lausanne (EPFL), University of Lausanne (UNIL), 1015 Lausanne, Switzerland; ines.villalba@epfl.ch (M.I.V.); sandor.kasas@epfl.ch (S.K.); 4Centre Universitaire Romand de Médecine Légale, UFAM, University of Lausanne, 1015 Lausanne, Switzerland; 5Department of Forensic Medicine, Institute of Further Training and Retraining of the Personnel, State Forensic Examination Committee of the Republic of Belarus, 220033 Minsk, Belarus; andrei.krilou@gmail.com; 6Department of Histology, Cytology and Embryology, Gomel State Medical University, 246000 Gomel, Belarus; nadyrov19621@gmail.com

**Keywords:** elastic modulus, viscoelasticity, nanomotion, actin cytoskeleton, cancer cell, BT-20 cell, ZR-75 cell

## Abstract

Cells of two molecular genetic types of breast cancer—hormone-dependent breast cancer (ZR-75 cell line) and triple-negative breast cancer (BT-20 cell line)—were studied using atomic force microscopy and an optical nanomotion detection method. Using the Peak Force QNM and Force Volume AFM modes, we revealed the unique patterns of the dependence of Young’s modulus on the indentation depth for two cancer cell lines that correlate with the features of the spatial organization of the actin cytoskeleton. Within a 200–300 nm layer just under the cell membrane, BT-20 cells are stiffer than ZR-75 cells, whereas in deeper cell regions, Young’s modulus of ZR-75 cells exceeds that of BT-20 cells. Two cancer cell lines also displayed a difference in cell nanomotion dynamics upon exposure to cytochalasin D, a potent actin polymerization inhibitor. The drug strongly modified the nanomotion pattern of BT-20 cells, whereas it had almost no effect on the ZR-75 cells. We are confident that nanomotion monitoring and measurement of the stiffness of cancer cells at various indentation depths deserve further studies to obtain effective predictive parameters for use in clinical practice.

## 1. Introduction

Breast cancer is the most prevalent cancer among women worldwide, making it a significant public health concern. Molecular biological sub-types of breast cancer are traditionally classified by the level of expression of the estrogen receptor (ER), progesterone receptor (PR), and human epithelial receptor 2 (HER2). About 80% of all cases of breast cancer are “ER-positive”. It means that cancer cells grow in response to the estrogen hormone. About 65% of them are also “PR-positive”. They grow in response to another hormone, progesterone. In about 20% of breast cancer cases, cells produce too many proteins known as HER2. These cancers tend to be aggressive and fast-growing. Some types of breast cancer (10–20%) are known as “triple negative breast cancer” because they do not have estrogen and progesterone receptors and do not overexpress the HER2 protein [1,2]. 

The mechanical behavior of cancer cells is closely related to cancer metastatic activity and disease progression, and is crucial for oncogenesis [3,4]. Cells are multi-level dynamic systems with highly integrated structure and function. The mechanical properties of cells allow them to withstand the various mechanical stresses in their environment and respond to stimuli of different origins and nature. The cell mechanical properties determine the biological behavior of cells: cell growth, differentiation, spreading, migration, motility, and death programs. Cancer metastasis is a complex process involving such stages as invasion into surrounding tissue, intravasation, transit in the blood or lymph, extravasation, and growth at a new site. All the stages involve mechanisms with the actin cytoskeleton participation [5,6]. 

The mechanical characteristics of cells depend on the mechanical properties of cell components, mainly cytoskeletal structures. The cell cortex is essentially responsible for the mechanical behavior of cells. It contains a plasma membrane-associated part of the actin cytoskeleton [7,8,9] and plays a crucial role in various cellular events: cellular mobility, differentiation, vesicular transport, proliferation, and cell death [10]. The structure and state of the actin cytoskeleton are modified during cancerogenesis, and abundant evidence indicates that cells from different cancers and stages differ in their mechanical properties. Various methods are nowadays available to evaluate the mechanical properties of single cancer cells [11,12,13,14,15,16,17,18,19,20]. Atomic force microscopy (AFM) is one of the most effective tools to characterize the mechanical properties of healthy and cancerous cells. The elastic modulus (Young’s modulus) is the most frequently used parameter to assess cellular mechanical properties. This parameter can be obtained by operating the AFM in such modes as force spectroscopy, force volume, Peak Force QNM, and others [21,22]. 

The cytoskeleton not only determines the shape and the mechanical properties of cells but also their dynamic behavior. Here again, modifications in cellular dynamics have been evidenced in cancer cells. Several studies have highlighted changes in the dynamic behavior of cancer cells after treatment with toxins that disrupt the cytoskeleton structure [23,24,25,26]. More recently, a peculiar type of dynamic behavior was highlighted in practically all living organisms, and was referred to as nanomotion [27,28]. This nanometric scale oscillation persists as long as the cells are alive and immediately stops when the cells die. Among other applications, nanomotion monitoring permits to conduct the rapid antibiotic antifungal [29] and antimitotic sensitivity tests [30]. These measurements are accomplished by monitoring the oscillations as a function of the time and the various drugs added to the analysis chamber. Cellular nanomotion was highlighted for the first time by attaching living cells to an AFM cantilever and monitoring its oscillations. More recently, our team demonstrated that traditional optical microscopes equipped with a video camera and dedicated software can also detect bacterial [31], fungal [32,33], and mitochondrial [34] nanomotion. This type of nanomotion detection is referred to as optical nanomotion detection (ONMD) [32]. 

In the present work, we focus on the analysis of the differences in the mechanical properties and behavior of breast cancer cells of two molecular genetic sub-types using different modes of AFM and ONMD methods. We used the ZR-75 and BT-20 breast cancer cell lines. According to a recent classification of breast cancer cell lines [1], the ZR-75 cell line is a sub-type A of luminous breast cancer, having receptors to estrogen (progesterone) and no receptors to HER2. The ZR-75 cells are characterized by a greater extent of cell differentiation, well-developed cytoskeleton structures, and a dense distribution of intercellular compounds. The BT-20 cell line is representative of triple-negative breast cancer of sub-type A, as it has no receptors for any of the compounds mentioned above. These cells are enriched with basal markers, including cytokeratin, indicating a lack of vimentin structures. Using three nanomechanical mapping modes of the AFM and ONMD methods, we have shown that two breast cancer cell lines differ in the spatial distribution of the actin cytoskeleton that strictly correlates with the spatial distribution of the parameters of cell elastic and viscoelastic properties and the parameters of the nanomotion of the studied cells. Since both cell lines have a different pathological impact, these experiments strongly suggest that the nanomechanical properties and nanomotion of single cancer cells could be linked to clinically relevant parameters. Our opinion is that this experimental field warrants greater attention and has the potential to yield concrete clinical applications in the future. 

## 2. Materials and Methods

Cell culture

Human breast cancer cell lines ZR-75 and BT-20 were obtained from the Republican Scientific and Practical Center of Oncology and Medical Radiology named after N.N. Alexandrov. For ONMD experiments, the cell lines BT-20 and ZR-75 were purchased from CLS cell lines service GmbH (Eppelheim, Germany). ZR-75 was cultured in RPMI-1640 (RPMI-A, Capricorn Scientific, Ebsdorfergrund, Germany) medium containing 10% fetal bovine serum and 1% penicillin–streptomycin solution at 37 °C (5% CO_2_), in Petri dishes and standard-coated T-75 vials (Sarstedt, Nümbrecht, Germany). BT-20 was cultured in DMEM/Ham’s F-12 (DMEM-12-A, Capricorn Scientific, Ebsdorfergrund, Germany) medium containing 10% fetal bovine serum (26140079, Gibco, Billings, MT, USA) and a mixture of antibiotics (100 U/mL penicillin; 100 g/mL streptomycin) in Petri dishes and T-75 adhesive-coated vials (Sarstedt, Nümbrecht, Germany). For the ONMD experiments, for BT-20 cell cultivation, 1% penicillin–streptomycin solution was used. A 5 min incubation with Trypsin-EDTA 0.25% (Gibco-Life technologies, Billings, MT, USA) at 37 °C (5% CO_2_) was used for dissociating adherent cells from the surface. Cells were transferred to Petri dishes with an adhesive coating and incubated for 24 h. For AFM scanning in air, the cells were fixed with 2% glutaraldehyde (20 min, 37 °C), washed with phosphate-buffered saline (PBS) and distilled water, and dried at room temperature in a laminar box.

Inhibition of actin polymerization was performed using a solution of cytochalasin D (22144-77-0, Sigma Aldrich, St. Louis, MO, USA) at a concentration of 1 mg/mL in DMSO for 10, 20, and 30 min, followed by washing with PBS (37 °C).

Fluorescence microscopy

The spatial actin cytoskeleton organization in cells of two cancer lines was analyzed using the labeling of polymerized actin. Cells were transferred to Petri dishes with a standard/adhesive coating, incubated for 24 h, and fixed with 3.7% formaldehyde for 10 min at room temperature. Cells were washed two times with PBS, permeabilized with 0.1% Triton X-100 (Sigma-Aldrich, St. Louis, MO, USA) for 5 min, and washed two times with PBS. For F-actin labeling, cells were incubated with Alexa Fluor 488 Phalloidin (A12379 Invitrogen, Waltham, MA, USA) using a 1:40 dilution in PBS for 20 min (at room temperature) followed by two additional washes. Subsequently, the cell DNA was stained with 4′,6-diamino-2-phenylindole (DAPI) (ab104139, Abcam, Cambridge, UK). The fluorescence of the cell samples was analyzed by an inverted light microscope (Axio Observer 3, Zeiss, Jena, Germany) with a solid-state light source Colibri 7 (Zeiss, Jena, Germany) and objective LD Plan-Neofluar 40x./0.6 Korr Ph2 M27. The image resolution was 2048 × 2048 pixels. Excitation and emission wavelengths were as follows: λ_ex_ = 353 nm and λ_em_ = 465 nm for DAPI; λ_ex_ = 493 nm and λ_em_ = 517 nm for Alexa Fluor 488 Phalloidin. The Zen Blue 2.5 Pro program was used to analyze the images.

ONMD method

Optical nanomotion studies (Figure 1a) were conducted on both adherent and suspended cells of BT20 and ZR-75 cell lines. The cells were cultivated following the protocol described in the cell culture section. The measurements of suspended cells were accomplished in uncoated IBIDI µ-Dish 35 mm, low (IBIDI 80131, Gewerbehof, Gräfelfing, Germany), which served as an analysis chamber. Absence of adhesion was obtained by coating the analysis chamber with a 75 µL layer of HFE 7500 fluorinated oil (3MTM, Novec™, Manassas, VA, USA) and allowing it to evaporate for five minutes. In this case, none of the cancer cells could attach to the glass surface of the analysis chamber, and cells adopted a spherical shape as depicted in Figure 1b,c,f,g. For adherent ones, the cells were seeded into 35 mm glass-bottomed dishes (μ-Dish 35 mm, high Glass Bottom, IBIDI Gewerbehof, Gräfelfing, Germany). Following a wash in PBS, the medium was replaced after 48 h. In this case, the cells could spread on the bottom of the analysis chamber, as depicted in Figure 1d,e,h,i, and adopted more polygonal and irregular shapes. The ONMD measurement consisted of recording 10 s movies of the living cells and their processing with a custom-made Matlab program. We used a Zeiss Axio Observer 7 microscope (Zeiss, Jena, Germany) operating with a 40× objective and equipped with a sCMOS Camera pco.edge Singapore). The microscope was equipped with a thermal system, which enabled the experiment to be conducted at 37 °C. The movies of cancer cell nanomotion were recorded at a frame rate of 30 fps at 0 and 30 min. The drug exposure measurements were conducted with 10 μM Cytochalasin D added to a Petri dish.

In that case, the cells could spread on the bottom of the analysis chamber (Figure 1) and adopt more polygonal and irregular shapes.

ONMD data analysis

A Matlab (v2023) custom-made software was used to process ONMD movies. After starting the program, the user is asked to isolate on the screen the cells to be analyzed (ROI (regions of interest)). The program eventually calculates the number of pixels that changed their intensity between frames separated by an adjustable number of images, referred herewith to as deltaF. A schematic of the analysis process is depicted in Figure 1a with a deltaF = 2.

The final result consists of a false-color image having its pixel intensities reflecting the movements that occurred during the recorded period. To get rid of possible “contaminants” such as small moving objects present in the field of view, only the user-selected regions (ROIs) are processed. The sum of the pixels composing the final image (Result image in Figure 1a) is referred herewith to as total movement. Its value is given in arbitrary units.

The data are represented as boxplots depicting the minimum, maximum, median, and first and third quartiles. The Mann–Whitney U test was used to compare the sample datasets.

Atomic force microscopy

Atomic force microscope Bruker BioScope Resolve (Bruker, Billerica, MA, USA) was used to study the mechanical and structural properties of the cell surface. In our experiments, we operated AFM in two modes: PeakForce Quantitative Nanomechanical Mapping (PF QNM) mode and Force Volume (FV) mode. Scanning in air was performed using a SCANASYST-AIR probe (Bruker, Camarillo, CA, USA, k = 0.4 N/m, R = 2 nm) in MIROview PF QNM mode. Whole cells were scanned with the scan size of 100 µm × 100 µm and a resolution of 512 × 512 pixels. The small areas of the cell surface (1 μm × 1 μm, 256 × 256 pixels) were scanned over the cell nuclear zones. The scan rate was 0.3 Hz, and the Peak Force Setpoint was 290 pN. To study the properties of live cells in liquid mode, the PF QNM in fluid mode, and FV in fluid mode were applied at 37 °C. The silicon tip with a nitride coating on a SCANASYST-FLUID (Bruker, Camarillo, CA, USA) cantilever was used (k = 0.7 N/m, R = 20 nm) for PF QNM in fluid. The scanning parameters were as follows: the scan size was 100 μm × 100 μm, resolution was 512 × 512 pixels, Peak Setpoint was 5 nN (for BT-20 cells) and 3.2–4.7 nN (for some experiments, 435 pN) (for ZR-75 cells). When FV was applied, a pre-calibrated silicone probe PFQNM-LC-A-CAL (Bruker, Camarillo, CA, USA) (k = 0.1 N/m, R = 70 nm) was used. The scanning parameters were as follows: scan size was 10 μm × 10 μm, the rate was 8.14 μm/s, and the trigger force threshold was 1 nN. The SCANASYST-AIR and SCANASYST-FLUID probes were calibrated before scanning the cell samples using the Touch mode, and the pre-calibrated PFQNM-LC-A-CAL probe was calibrated using the No-Touch mode.

AFM data analysis

AFM data obtained in PF QNM mode were analyzed using NanoScope Analysis 1.9 software (Bruker, Billerica, MA, USA). Three AFM images recorded simultaneously in PF QNM mode were included in the analysis: topography (height), maps of elastic modulus (DMT modulus), and maps of the indentation depth (indentation). To estimate the elastic modulus (Young’s modulus), the linear segment of a retract curve recorded during scanning was fitted automatically using the Derjaguin–Muller–Toporov (DMT) model. The raw AFM data from files generated by the Bruker instrument in the process of the scanning of the cell surface were transformed into ASCII files for their further processing.

AFM data obtained in FV mode were analyzed using NanoScope Analysis 1.9 software and the custom-written code in Matlab that was kindly shared with us by Dr. Efremov. The Matlab script allows for the extraction of the parameters of viscoelastic properties from the standard force–distance curves obtained using FV mode. The method is based on the theoretical model developed by Ting [35] for the problem of the indentation of a linear viscoelastic half-space by a rigid axisymmetric indenter for arbitrary load history [36,37]. Earlier, the method was effectively applied to measure the viscoelastic properties of several benign and cancerous cell lines. The following parameters of viscoelasticity were calculated using the power-law rheology (PLR) model (also known as a springpot in parallel with a dashpot): the instantaneous elastic modulus E_0_ and the power-law exponent α.

The statistical analysis of experimental data was performed using OriginPro, version 2019b, Statistical calculator “Statistics Kingdom” (https://www.statskingdom.com/, accessed on 20 September 2023) and Rstudio (version 4.2.1) packages (ggplot2,tidyverse,dplyr,plyr). The data were checked for compliance with the normal distribution law using the Shapiro–Wilk test. The data are represented as either the median and limits of the interquartile range (Me(LQ; UQ)) or the mean and the limits of 95% confidence interval (95% CI)/or the mean and the standard deviation (M ± SD). The Mann–Whitney U test or *t*-test was used to compare the sample parameters.

## 3. Results

### 3.1. Morphology and the Actin Cytoskeleton Organization of ZR-75 and BT-20 Cells

Figure 2 represents the AFM topographic images of the live studied cells and the spatial distribution of the actin cytoskeleton density in the cells. Two cancer cell lines differ sharply in the actin cytoskeleton organization. BT-20 cells have a cortical layer of high-density actin cytoskeleton. In the center of BT-20 cells, the density of actin cytoskeleton structures is significantly reduced. ZR-75 cells are characterized by a more monotonous spatial distribution of actin cytoskeleton. Having a dense layer of actin cytoskeleton in the cortex, the cells also possess relatively dense actin cytoskeleton structures in all of the cell body. Under the same Alexa Fluor488-Phalloidin fluorescence recording conditions, the fluorescence intensity at 4–5 μm from the cell edge (the maximum of F-actin density) for BT-20 cells have F-actin density about 7–10% higher than ZR-75 cells have (11,971 ± 173 rel. units compared to 11,192 ± 58 rel. units, *p* < 10^−6^, *t*-test (pooled variance)). At the distance of 19–20 μm from the cell edge, the F-actin density for BT-20 cells was 22% lower than one for ZR-75 cells (8055 ± 28 rel. units compared to 10,362 ± 35 rel. units, *p* < 10^−6^, *t*-test (pooled variance)).

The actin cytoskeleton of BT-20 cancer cells is highly sensitive to the cytochalasin D. The effect of this potent inhibitor of actin polymerization and actin cytoskeleton structure disturber is quicker than its effect on the actin cytoskeleton of ZR-75 cells (Figure 3a–h). So, the F-actin density peak in the cortex disappears at 20 min after cytochalasin D is added to BT-20 cancer cells. For ZR-75 cancer cells, this peak is visible up to 30 min incubation time with cytochalasin D (Figure 3i,j).

These features of the actin cytoskeleton organization of two cancer cell lines seem to determine most of the distinguishing features of their morphology. Most BT-20 cells in the cell samples were located as cell groups or as separately lying cells and had oval and polygonal shapes. Up to 30% of the BT-20 cells were spindle-shaped with long processes. Mature cells of oval and polygonal form had a norm-chromic nucleus with 2 to 5 nucleoli. The cytoplasm had a well-defined perinuclear zone with organelles. The free surface of the cells had short microvilli. There were more microvilli on the contact surfaces between cells, and they were longer. The ZR-75 cells formed a monolayer in the cell samples with up to 20% of single-layered cells. The ZR-75 cells were bigger (contacting area and height) compared to the sizes of BT-20 cells. The cells had long processes, hypochromic nuclei with 4 to 10 nucleoli, and the cytoplasm defined a perinuclear zone of the organelles well.

### 3.2. Elastic Properties of ZR-75 and BT-20 Cells Measured by Force Volume and Peak Force QNM Modes of AFM

We studied the mechanical properties of BT-20 and ZR-75 breast cancer cells using three modes of AFM (Figure 4). Using the first method, we fixed the cell structure with glutaraldehyde and dried cell samples and then recorded the AFM images using PF QNM in air. The PF QNM in air mode provides an excellent opportunity for imaging nanoscale features of the cell surface morphology (Figure 4a,d) and mapping the mechanical properties of small cell surface areas with nanoscale resolution (Figure 4g,j). The size of a pixel in nm for measuring the mechanical parameters was about 4 nm. Using two other modes, we studied the cell in a live state in liquid. The second method also belongs to the types of the PF QNM mode. Because of work in liquid, this method provides data on the cell properties with a lower extent of spatial resolution. PF QNM in fluid helps observe the features of the surface and cortex of cells in a live state (Figure 4b,e). The size of a pixel in nm for measuring the mechanical parameters was about 190–200 nm (Figure 3h,j). The third method was FV in fluid mode. The FV mode differs principally from the PF QNM mode in the manner of mechanical indentation and image recording parameters. This method was not used for analyzing the cell morphology (Figure 4c,f), only for measuring the elastic properties of the cell surface (Figure 4i,l). The size of a pixel in nm for determining the mechanical parameter was about 500 nm. The AFM probe in two modes of AFM (PF QNM and FV modes) indents the cell surface at different depths. Figure 5 presents the dependencies of the elastic modulus obtained using two different AFM modes on the indentation depth. The maximal indentation depth (achieved automatically) of the PF QNM images of the cell surface obtained for live cells was about 300 nm. For force-distance curves of the FV images, the indentation depth was significantly higher (even up to 1700–2000 nm). Figure 5 shows that the elastic modulus decreases with the indentation depth for both cancer cell lines in both ranges: lower 300 nm (PF QNM mode) and higher 300 nm (FV mode). The obtained values of the elastic modulus for two indentation ranges presented in Figure 5 are different, mainly due to the difference in the indenting parameters of the different AFM modes. We also used the DMT model for determining the elastic modulus when using the PF QNM mode and Hertz’s model in the case of the FV mode. Understanding the features of data obtained using the AFM modes, we can image virtually the combined dependencies of the cell elastic parameter on the indentation depth. At the small indentation depth up to 200–300 nm, the elastic modulus of BT-20 breast cancer cells is more than that of ZR-75 breast cancer cells. So, in liquid, for live cells at 120 nm indentation depth, the elastic modulus obtained using PF QNM was 888.3(603.5;1153.2) kPa for BT-20 cells and 396.9(188.8;585.4) kPa for ZR-75 cells (the indentation depth was 114.8(104.8;130.0) nm for BT-20 cells and 131.9(121.1;138.9) nm for ZR-75 cells).

For the comparison, at 20 nm indentation depth, the elastic modulus obtained using PF QNM was 32.56(26.40;42.28) MPa for BT-20 cells and 1.47(1.05;4.31) MPa for ZR-75 cells (the indentation depth was 17.3(15.4;18.7) nm for BT-20 cells and 18.5(15.3;19.6) nm for ZR-75 cells).

The higher stiffness of the BT-20 cells compared to the stiffness of ZR-75 cells at the small indentation depth is proved by the data obtained in the experiments with glutaraldehyde-fixed and dried cells. In air at 20 nm indentation depth, the elastic modulus obtained using PF QNM in air was 64.83(61.99;68.27) MPa for BT-20 cells and 45.20(42.75;49.70) MPa for ZR-75 cells (the indentation depth was 25.1(23.7;25.8) nm for BT-20 cells and 19.2(16.9;20.1) nm for ZR-75 cells).

With increasing the indentation depth, the elastic modulus of BT-20 cells becomes lower than the modulus of ZR-75 cells. So, at 350 nm indentation depth, Young’s modulus obtained using FV mode in liquid was 1.68(0.90;3.12) kPa for BT-20 cells and 4.81(4.35;5.38) kPa for ZR-75 cells and at 950 nm was 1.06(0.76;1.48) kPa for BT-20 cells and 2.16(1.90;2.35) kPa for ZR-75 cells.

The obtained data show the presence of the stiff cortical layer with a relatively soft central part in BT-20 cells. The stiffness of the ZR-75 cells gradually decreases with a distance from the edge to the center of the cells. The actin cytoskeleton (being principally responsible for the cell stiffness) is distributed non-homogeneously throughout the cell. As shown in Figure 2b,d and Figure 3i,j its density is maximal in the cortical layer (under the plasma lemma), corresponding to the data on the dependence of the cell elastic modulus on the indentation depth obtained.

### 3.3. Viscoelastic Properties of BT-20 and ZR-75 Breast Cancer Cells Measured Using FV AFM Mode

The response of cells as other biological materials is dependent upon how quickly the mechanical load is applied or removed, and the extent of deformation is dependent upon the rate at which the deformation-causing loads are applied. The time-dependent material behavior called viscoelasticity is the essential property of cells. The FV mode of AFM involving point-by-point acquisition of the force curves over the sample area is frequently used for measuring the viscoelastic properties of materials. To analyze our FV data, we used the power-law rheology (PLR) model, which provides good results in terms of describing the relaxation behavior of cells [37]. The PLR model is characterized by two parameters: the instantaneous elastic modulus E_0_ and the power law exponent α. Because a high α value is related to a high extent of relaxation, materials exhibit a solid-like behavior at α = 0, and a fluid-like behavior at α = 1 [36]. Figure 6 represents the viscoelastic parameters E_0_ and Young’s modulus (E_Hertz_) that were also extracted from the force–distance curves recorded using the FV mode for two types of breast cancer cell lines.

The parameters in Figure 6 are analyzed for two indentation depths: about 350 nm and 950 nm. According to Hertz’s approach, Young’s modulus of BT-20 cells is significantly lower than that for ZR-75 cells at two indentation depths. At the same time, the instantaneous modulus of BT-20 cells at 350 nm indentation depth is higher, and at 950 nm indentation depth, it is lower than one of ZR-75 cells. The power-law exponent of BT-20 cells at both used indentation depths is higher than that of the ZR-75 cells. Moreover, with increasing the indentation depth, the instantaneous modulus decreases, and the power-law exponent of both cell lines increases.

### 3.4. Nanomotion Parameters for BT-20 and ZR-75 Breast Cancer Cells

The main aim of the nanomotion experiments was to explore the potential of the ONMD method in monitoring the effects of an actin cytoskeleton-depolymerizing drug, cytochalasin D, on BT-20 and ZR-75 cancer cells. Figure 7 shows the evolution of the nanomotion signal in the studied cancer cells (its increase or decrease) before and after the injection of cytochalasin D into the cell cultures.

Control value was obtained by averaging the nanomotion amplitude of every single cell between 0 and 30 min before the addition of cytochalasin D. The cytochalasin D data correspond to the averaged nanomotion amplitude of single cells between 0 and 30 min after cytochalasin D injection. The effects of cytochalasin D are significant in the BT-20 cell nanomotion, showing a decrease in the nanomotion parameters for the cell suspended state and their increase for its adherent state. In the case of ZR-75 cells in a suspension, significant differences are also observed. However, for ZR-75 cells in the adherent state, no detectable modification occurs after cytochalasin D injection, except for higher variation in the values following the effects of the drugs. However, the nanomotion parameter of ZR-75 cells in the adherent state is significantly higher than that of the BT-20 cells in the absence of the drug. In the control experiments, no significant difference was found between the suspended and adherent states for any of the cell lines in terms of ONM evolution (*p* > 0.05, Mann–Whitney U test).

## 4. Discussion

Testing cancer cells’ nanomechanical parameters such as cell elasticity or its nanomotion constitutes one of the perspective directions for the development of novel technologies for early cancer diagnosis and effective anticancer therapy. Information about the cell’s mechanical properties (i.e., their Young’s modulus) can be obtained through the analysis of force–distance curves. These curves are obtained by indenting (pushing) the AFM tip into the cell and monitoring the cantilever deflection during the indentation process. Several previous studies explored the elastic modulus of breast cancer cells using the AFM (Table A1). They revealed that the cancer cell’s Young’s modulus is characterized by a wide range of values (0.3–30 kPa) and depends on several factors. Among those, we should mention the AFM mode used to obtain the indentation curves (Force Spectroscopy (FS), Force Volume (FV), Peak Force (PF) or Quantitative Imaging (QI)) [12,38,39,40,41,42,43,44], the scanning rate, the applied loads, the tip motion pattern, its geometry, the theoretical model to calculate the cellular Young’s modulus, and the cancer cell line used in the study. Additional parameters such as the cellular state (dividing or not), the explored region (above the nucleus or in its vicinity), the substrate stiffness, the imaging medium temperature, composition conditions, and cellular density play also important roles in the measurements [11,17,45,46,47,48,49,50]. Thus, the elastic modulus value depends on many parameters and conditions that can only be partially met in studies aiming to reproduce previously published data. However, despite these difficulties, we are convinced that exploring the heterogeneity of the elastic properties of the different cancer cells is worth studying in detail since it could lead to valuable clinically significant parameters [18].

In the present work, we focused our attention on the spatial distribution of the elastic modulus of two different breast cancer cell lines (estrogen-sensible, ZR-75; and triple-negative, BT-20). We used three different standardized approaches to evaluate the elastic modulus. Two of them concerned living cells in liquid under physiological conditions (PF QNM in fluid, FV). The third approach involved chemically fixed and dried cell samples (PF QNM in air). The chemical fixation of cell structures and further hydration are widespread preparation methods in histology and cytology that are also effectively used to determine a difference in properties (morphology) of cancer cells. It should be kept in mind that cells are viscoelastic structures and that their mechanical properties therefore depend on the time scale of the measurement (i.e., indentation speed). At a shorter time scale, the AFM signal mainly comes from the elastic response of the cells. At a longer timescale, the sample creeps, and the AFM signal essentially contains the relaxation component. The time scale of AFM indentation corresponds to the tip velocity. Hence, data comparison can only be achieved when the force curves are acquired with the same indentation velocity. Similarly, we only can compare the data if the measurements were made using the same AFM operating mode.

We employed the widely used Hertz model to process our experimental data. It infers the force–indentation relationship for infinitesimal indentations of purely elastic materials by axisymmetric indenters. While satisfying the assumptions of applicability of Hertz’s theoretical model in the case of rigid inorganic materials, the different methods like PF QNM and FV showed Young’s moduli with similar numerical values [51]. Some of the assumptions of Hertz’s model do not apply to cell indentation. Hertz’s model assumes homogeneous and linearly elastic material, while cells are heterogeneous and non-linearly elastic materials. To maintain linearity, it is recommended to evaluate the elastic modulus at a small indentation depth. The choice of the correct indentation depth is related to the size of contact of the AFM probe tip to the cell body. Using a commercially sharp AFM probe inevitably leads to a nonlinear stress–strain response of the sample material [52,53]. On the other side, sharp probes allow for the acquisition of high-resolution cell topography data simultaneously with the map of cell mechanical properties, which is not feasible in cases of the usage of big radius blunt and spherical probes. In many cases, Young’s modulus maps are presented in the literature without mentioning the indentation depth, which seems to be meaningless [54].

Cells are complex structures with clear centers (the nucleus) and periphery (the plasmalemma). The basic cell organelles and other cell structures are typically located along the axis of the plasma membrane and nucleus, reflecting the directions of signal pathways in and out of cells. Therefore, intuitively, heterogeneous cell material can be imaged as a layered body with separate conditional layers located at a certain distance along the axis of the plasma membrane and nucleus. In principle, it can be assumed that within these thin layers of the thickness of
(1)$$\Delta \delta_{i}=\delta_{i+1}-\delta_{i},$$
where $\delta_{i}$ is the distance between the external cell edge and the beginning (first limit) of the considered cytoplasmic i-layer, and $\delta_{i+1}$ is a distance between the external cell edge and the second limit of the considered cytoplasmic layer, the mechanical properties do not change significantly (the elastic modulus is $E_{i}$). Thus, we can assume that the dependence of the cell’s mechanical properties on the indentation depth
(2)$$E_{i}=f\left(\delta_{i}\right)$$
reflects the peculiarity of the spatial organization of cells, mainly their cytoskeleton structure. This dependence may be a unique cancer cell characteristic.

Our data obtained using the different AFM-based approaches suggest a decrease in the elastic modulus with the indentation depth. This decrease in the elastic modulus with indentation is observed for cells in both live and dried states [55]. There is evidence in the literature about the mechanism of pseudo-softening of the material with the indentation depth [54]. Because of the pseudo-softening, the decrease in the elastic modulus of live cells with the indentation depth from 200 to 1000 nm does not exceed 8–9%. The changes we have obtained in Young’s modulus of cancer cells within the same indentation depth limits are much higher.

It is necessary to understand that going beyond the conditions of the applicability of the basic Hertz model, we enter a zone of uncertainty in the evaluation of the absolute value of Young’s modulus. In our experiments, the radius of the AFM probe tip used in each selected method exceeds the indentation depth. However, establishing a high coefficient of correspondence between the experimentally obtained force–distance curves and theoretical curves allows us to speak about the apparent elastic modulus (apparent Young’s modulus) and use it as a characteristic of the structure and properties of cancer cells. Moreover, the characters of the dependencies are similar when using independent AFM modes. The maximal indentation depth for each method used in the study did not exceed the recommended 10–20% of the cell height at the point of indentation. One more problem in the cell modulus assessment within a wide range of the indentation depth is a switch between 2D and 3D mechanics. At the indentation depth of the order of the magnitude of the cortex mesh, the AFM probe indents mostly the cortical cytoskeleton, and the other cell components are affected by the tip load to a lesser extent. The cell surface resistance to the load is represented better by 2D physics. At the deeper indentation depth, in the contact between the AFM probe and cell surface, larger areas of the cytoskeleton and other components are involved, and the description of the bodies’ contact requires 3D theories [56].

The obtained dependencies of the elastic modulus on the indentation depth differ significantly for BT-20 and ZR-75 cells. Within the 200–300 nm cortical layer (the cortex mesh), the BT-20 breast cancer cells were much stiffer than ZR-75 breast cancer cells. Opposite to this, at the indentation depth larger than 300 nm, ZR-75 cells were stiffer than BT-20 cells. The cell cortical layer of about 300 nm thickness contains mainly cortical actin cytoskeleton structures. According to fluorescence microscopy data, the concentration of F-actin at the cell edge for BT-20 cells is much higher than that of ZR-75 cells. In the deeper layers of the BT-20 cells, a lack of F-actin structures supports the more viscous behavior of the cell material. The dense cortical actin cytoskeleton supports the formation of cytoplasmic processings (microvilli) at a high density and tight adhesion to the substrate and other cells [57]. The behavior of BT-20 cells seems to be similar to liquid bodies with a relatively rigid envelope. In this work, we focused on the spatial actin cytoskeleton structures in BT-20 and ZR-75 cancer cell lines because their difference may explain different characters of the dependence of Young’s modulus on the indentation depth for the cells. The actin cytoskeleton makes a major contribution to cell biomechanical parameters measured by ACM [58]. The effect of microtubule-targeting drugs is more difficult to detect by AFM. Due to their location in the depth of the cells, they play a less important role in the cellular mechanical properties as measured by AFM [58,59]. Two cancer cell lines differ in an intermediate cytoskeleton: ZR-75 cells have a vimentin cytoskeleton, and BT-20 cells have a cytokeratin cytoskeleton. However, the intermediate cytoskeleton spatial structure does not clearly explain a sharp change in the ratio of Young’s moduli of two cell lines within the 200–300 nm cell surface layer.

ZR-75 cells have actin cytoskeleton structures distributed more homogeneously within the cell body than that of BT-20 cells. It supports good adhesive properties and high motility of ZR-75 cells. In parallel to the growth of viscous properties with the deepness into ZR-75 cells, the elastic modulus of the cells decreases to a lesser extent compared to that in BT-20 cells. The reasons for such mechanical behavior of ZR-75 cells can be both a highly developed actin cytoskeleton structure and a dense vimentin cytoskeleton structure. The relative weakness in ZR-75 cell elastic properties at the cell edge and lower density of their cortical cytoskeleton structure compared to ones of BT-20 cells can evidence highly dynamic processes of actin cytoskeleton rearrangement at the cell periphery [60]. The ZR-75 cell line is known to possess a higher metastatic potential than BT-20. Interestingly, its “spontaneous” (control) ONMD signal evolution is higher than that of BT-20 in an adherent state, as depicted in Figure 7b. It indicates that the nanomotion amplitude of ZR-75 cells increases as a function of time when the cell is attached. This observation deserves further investigation by ONMD as well as by AFM-based nanomotion detection. The effect of cytochalasin D on attached BT-20 cells is stronger in terms of nanomotion than on the ZR-75 cell line. The higher concentration of actin in the cortical region in the BT-20 cell line compared to the ZR-75 cell line could be an explanation. It would indicate that the depolymerization of actin filaments induces an increase in the nanomotion activity, which is proportional to the actin concentration before the drug treatment.

On the one side, the differences in the actin cytoskeleton of cells of hormone-sensitive types of breast cancer and triple-negative breast cancer determine the difference in their ability to migrate, proliferate, and establish metastatic colonies. On the other side, understanding the cytoskeleton structure features of different molecular genetic sub-types of breast cancer allows for the development of new therapeutic methods specific to the cancer sub-types. Some new ruthenium(II) complexes with gallic acid synthesized recently showed a higher level of cytotoxicity and actin cytoskeleton rearrangement in triple-negative breast cancer cells (MDA-MB-231cells) that led to inhibition of invasion, migration, and adhesion, compared with the hormone-dependent breast cancer cells (MCF-7 cells) [61]. Our data and those of the literature indicate that the drugs that caused the actin filament depolarization can be considered effective agents against triple-negative breast cancer [62].

## 5. Conclusions

We have demonstrated in our work on the example of hormone-sensitive and triple-negative breast cancer cell lines (ZR-75 and BT-20 cell lines) that AFM and ONMD techniques are effective for measuring the parameters of cell nanomechanical behavior to form the mechanical phenotype of cancer cells of a selective molecular genetic cancer sub-type. The dependencies of elastic/viscoelastic properties on the indentation depth and the nanomotion signal on cell contact-to-substrate state and the presence of a drug causing the actin filament depolarization relate to the organization of actin cytoskeleton in cancer cells and present the cancer cell unique characteristics. These cell mechanical phenotype characteristics will help a deeper understanding of the cancer cell behavior in cancerogenesis and the development of effective therapeutic procedures against the specific sub-types of breast cancer, including personalized therapy.

## Figures and Tables

**Figure 1 cells-12-02362-f001:**
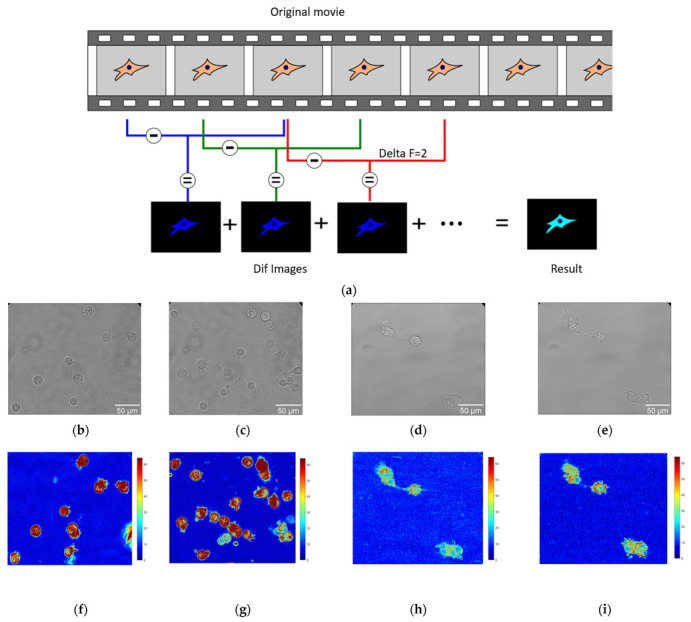
The application of the ONMD method to cell study. (**a**) Scheme of processing the movies using the ONMD algorithm. Every single frame is subtracted from the frame located deltaF frames later (in this case deltaF = 2). The absolute value of the obtained images (Dif images) is summed and produces the final false-color image (Result). The brighter it is, the more pixels change their value during the recording. (**b**–**e**) The optical images of suspended BT-20 cells (**b**,**c**) and adherent ZR-75 cells (**d**,**e**). (**f**–**i**) Corresponding to optical images (**b**–**e**), the false-color images of the number of pixels that changed their intensity between frames for suspended BT-20 cells (**f**,**g**) and adherent ZR-75 cells (**h**,**i**) (same magnification as the corresponding optical images (**b**–**e**)).

**Figure 2 cells-12-02362-f002:**
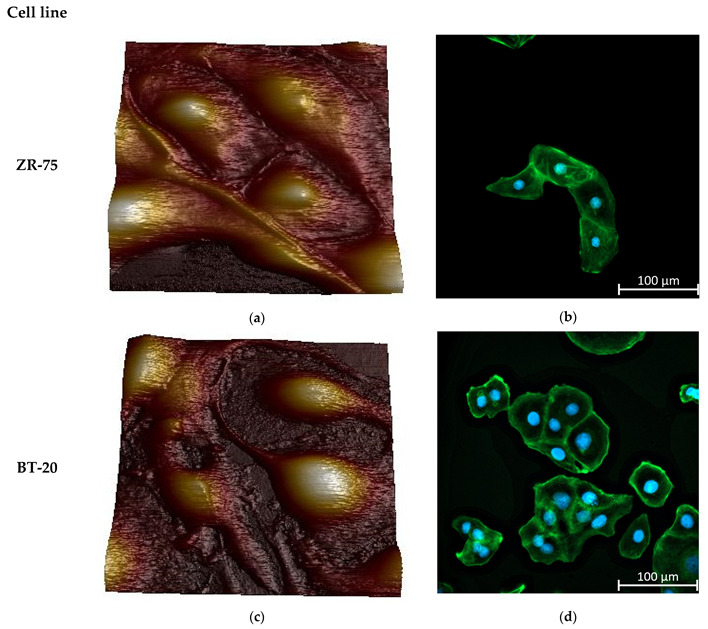
Morphology and actin cytoskeleton organization in BT-20 and ZR-75 breast cancer cells. (**a**) Three-dimensional topographic AFM image of living ZR-75 cells. The image was recorded using PF QNM in fluid mode. The scan size is 94.7 μm × 94.7 μm; the resolution is 521 × 512 pixels. (**b**) Representative immunofluorescence image showing the distribution of F-actin staining by Alexa Fluor488-Phalloidin in ZR-75 cells (green). Nuclei stained with DAPI are shown in blue. The exposure time for Alexa Fluor488-Phalloidin fluorescence was 1.5 s. The image size is 333 μm × 333 μm; the resolution is 2048 × 2048 pixels. (**c**) Three-dimensional topography AFM image of living BT-20 cells. The image was recorded using PF QNM in fluid mode. The scan size is 94.7 μm × 94.7 μm; the resolution is 521 × 512 pixels. (**d**) Representative immunofluorescence image showing the distribution of F-actin staining by Alexa Fluor488-Phalloidin in BT-20 cells (green). Nuclei stained with DAPI are shown in blue. The exposure time for Alexa Fluor488-Phalloidin fluorescence was 24 μs. The image size is 333 μm × 333 μm; the resolution is 2048 × 2048 pixels.

**Figure 3 cells-12-02362-f003:**
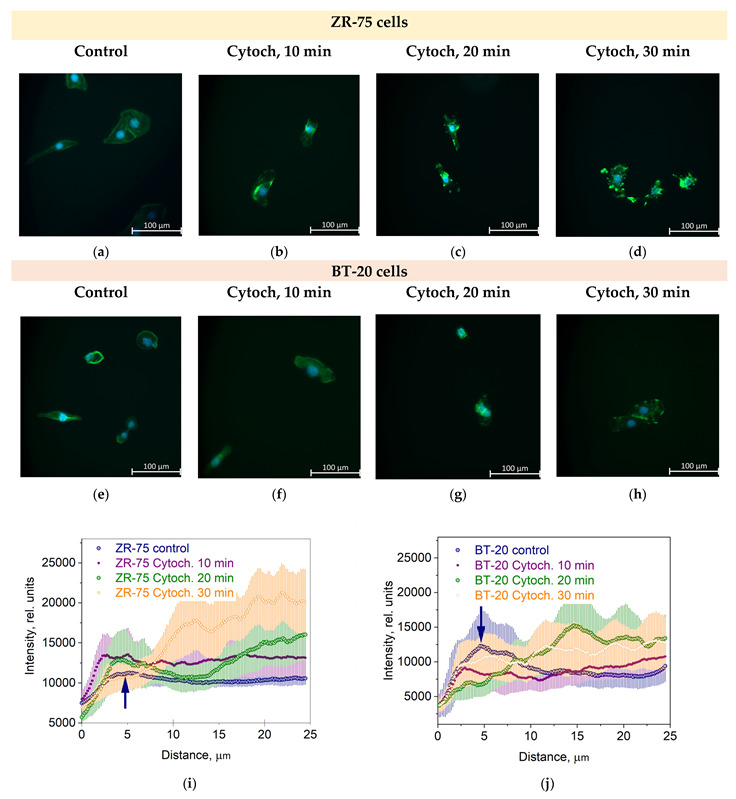
The effect of cytochalasin D (Cytoch.) on the spatial distribution of actin cytoskeleton structures in BT-20 and ZR-75 breast cancer cells. (**a**–**h**) Representative immunofluorescence images showing the distribution of F-actin staining by Alexa Fluor488-Phalloidin (green) in ZR-75 cells (**a**–**d**) and BT-20 cells (**e**–**h**) after cell treatment with cytochalasin D (10–30 min). Nuclei stained with DAPI are shown in blue. The exposure time for Alexa Fluor488-Phalloidin fluorescence was 1.5 s for every experiment. The image size is 333 μm × 333 μm; the resolution is 2048 × 2048 pixels. (**i**,**j**) The averaged profiles of the F-actin density spatial distribution in a direction from the cell edge to the nucleus after cell treatment with cytochalasin D (10–30 min). Data are presented as the mean and the limits of 95% CI. The blue arrows indicate the maximal density of the actin cortical cytoskeleton.

**Figure 4 cells-12-02362-f004:**
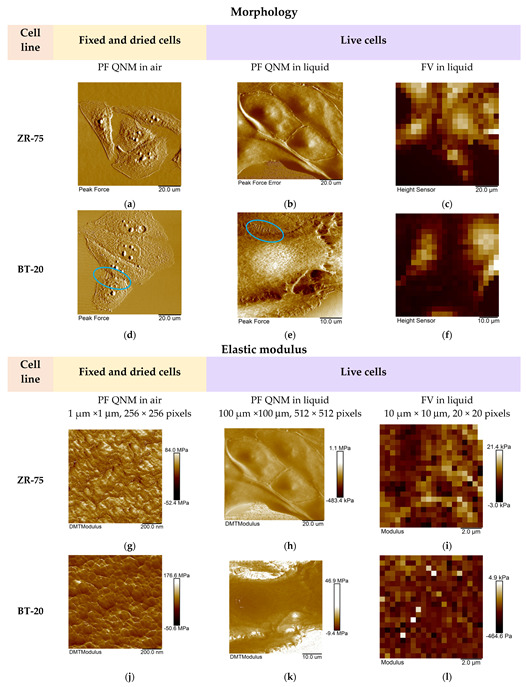
Representative AFM images of the studied breast cancer cells recorded using three AFM modes. (**a**–**f**) The AFM images (PeakForce, PeakForce Error, and Height) obtained using different AFM modes and representing the studied cell morphology. Blue ovals show the microvilli of BT-20 cells observed in the AFM images recorded using PF QNM mode in air and liquid. The size of the images recorded using FV mode was 10 μm × 10 μm. (**g−l**) The maps of the elastic modulus (DMT modulus and Young’s modulus) of the studied cells obtained using different AFM modes.

**Figure 5 cells-12-02362-f005:**
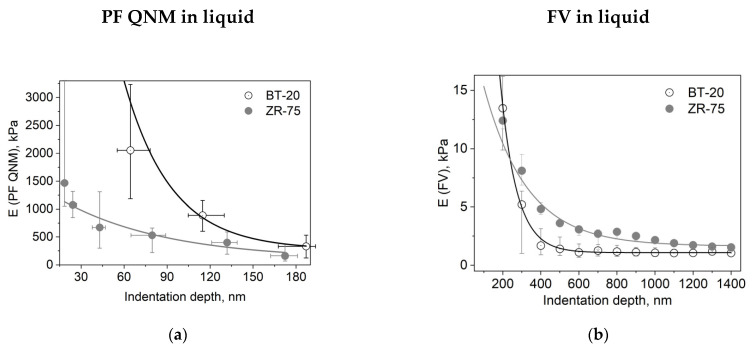
Dependencies of the elastic modulus (E) on the indentation depth for different cell types and AFM modes. (**a**) Dependence of the DMT modulus obtained using the PF QNM in fluid data on the indentation depth. The dependence was obtained using the data of channels DMT modulus and indentation. (**b**) Dependence of the Young’s modulus obtained using the FV data on the indentation depth. Data are presented as Me(LQ; UQ).

**Figure 6 cells-12-02362-f006:**
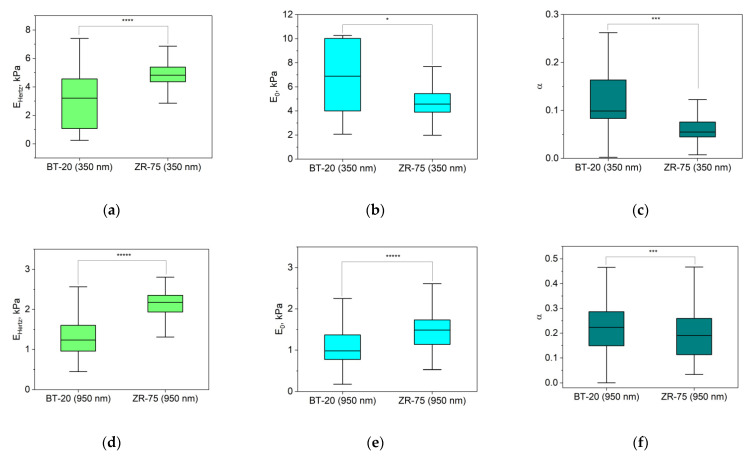
Elastic and viscoelastic parameters of BT-20 and ZR-75 cells. E_Hertz_ (**a**,**d**) is Young’s modulus isolated from the force–distance curves of FV data based on Hertz’s theory. The viscoelastic parameters (the instantaneous elastic modulus, E_0_ (**b**,**e**), and the power law exponent, α (**c**,**f**)) were extracted from force–distance curves of FV data based on the PLR theory. The parameters were measured at the indentation depth of 350 nm (376(333;393) nm for BT-20 cells and 356(331;377) nm for ZR-75 cells) and 950 nm (949(925;975) nm for BT-20 cells and 944(923;973) nm for ZR-75 cells). Data are presented as Me(LQ; UQ). * *p* < 0.05, *** *p* < 0.001, **** *p* < 0.0001, ***** *p* < 0.00001, Mann–Whitney U test.

**Figure 7 cells-12-02362-f007:**
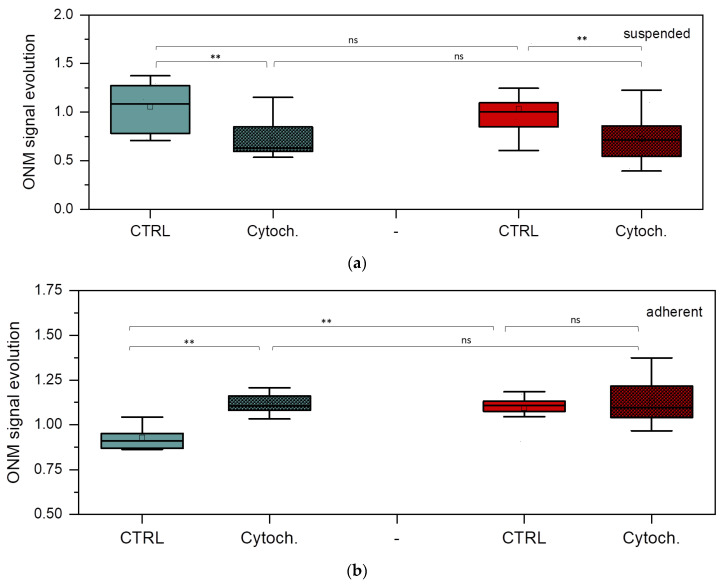
Nanomotion parameters of BT-20 and ZR-75 cells. ONM signal evolution of suspended (**a**) and adherent (**b**) BT-20 (green) and ZR-75 (red) cells after 30 min cell incubation without and with cytochalasin D. Mann–Whitney U test: ** *p* < 0.01, ns (not significant); the number of cells (n): suspended BT-20 cells n = 5 (control, ctrl) and n = 9 (cytochalasin D, cytoch.); adherent BT-20 cells n = 12 (ctrl) and n = 8 (cytoch.); suspended ZR-75 cells n = 14 (ctrl) and n = 7 (cytoch.); adherent ZR-75 cells n = 11 (ctrl) and n = 14 (cytoch).

## Data Availability

The original contributions presented in the study are included in the article, further inquiries can be directed to the corresponding author.

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
