# Peer review of "Mechanical Properties and Nanomotion of BT-20 and ZR-75 Breast Cancer Cells Studied by Atomic Force Microscopy and Optical Nanomotion Detection Method"

_cells, 2023, doi:10.3390/cells12192362_

Round 1

Reviewer 1 Report

Hello

The current research was very good, interesting and usefulØŒ The only questionØŒ is why the BT20 was used and Why was Mdamb231 not used? And isn't a normal  cell necessary?

Hello

The current research was very good, interesting and useful The only question is why the BT20 was used Why was Mdamb231 not used? And isn't a normal cell necessary?

Author Response

Q.:The current research was very good, interesting and usefulØŒThe only questionØŒ is why the BT20 was used and Why was Mdamb231 not used? And isn't a normal  cell necessary?

R.: Cell lines with a triple negative status of ER, PR, and HER2 are differentiated as basal A (more luminal-like) and basal B (more basal-like) cell lines. BT-20 is classified as basal A, and MDAMB231 is classified as basal B (Dai et al., 2017). The composition and structure of the cytoskeleton are different in these cells. According to Hu  (Hu et al., 2018), the cells of triple-negative breast cancer of MDAMB231 (type B) and MDAMB468 (type A) have different spatial structures of actin cytoskeleton, elastic modulus (shear modulus) and viscosity. Our task was to use a combined approach to study the difference in the mechanical behavior of hormone-sensitive and triple-negative breast cancer cell lines. We are sure there is a difference in the mechanical behavior of ZR75 and MDAMB231, and even BT20 and MDAMB231. We did not aim to compare the normal breast cells with cancer breast cells.

Reviewer 2 Report

In the manuscript “Mechanical properties and nanomotion of BT-20 and ZR-75 breast cancer cells studied by atomic force microscopy and optical nanomotion detection method” the authors apply an atomic force microscopy technique along with an optical motion detection method to analyze the mechanical properties of hormone-dependent and triple-negative breast cancer cell lines.

In their work authors demonstrate that the phenotype of cancer cells can be linked to the cells' nanomechanical behavior. The cell mechanical “portrait” could serve not only as a marker of malignancy but can help a deeper understanding of the cancer cell behaviour in the process of tumour formation.

Although the obtained results are very interesting and important, I feel that the paper needs some revision and clarification before it can be published.

My comments and questions to the authors are outlined below:

It is not clear to me why cells were fixed before staining with fluorescent dyes? I would expect to see the labelling process on the living cells.

I would recommend adding an ONMD control experiment on the fixed cells in liquid to exclude any external factors like vibration or setup noise.

Why the PF QNM setpoint is so different for air and liquid measurements? I would expect the setpoint in liquid to be smaller in comparison to the air.

I think there is a mistake and the FV threshold was not 1N (line 207).

Since the probe radius of the PFQNM-LC-A-CAL was only 70 nm, I don’t think that the Herz model applies to the modulus calculation. I would encourage the authors to use the Sneddon approach or a model for parabolic-shaped AFM probe.

The paper must be revised in the English language. I would recommend rewriting the abstract to make it more readable.

Author Response

Thank you very much for the comments. Please, see the attachment. 

Reviewer 3 Report

In this paper, authors measured the nanomechanical parameters of ZR-75 and BT-20 cells using atomic force microscopy and ONMD technology. This is an interesting study of cancer cell biomechanical profile. The results are conducive to a deeper understanding of the biomechanical behavior of cancer cells and may provide a new perspective for future cancer treatment programs.

However, the following issues should be addressed before publication.

In the results of AFM, it is suggested to provide the force-distance curves. The result showed that the dependencies of the elastic modulus on the indentation depth differ significantly for BT-20 and ZR-75 cells. Authors need to discuss the clinical significance in detail of these different biomechanical profiles.

In this study, the authors focused in role of actin cytoskeleton in the mechanical properties of two molecular genetic types of breast cancer cells. There is a possibility that other cytoskeleton (e.g. microtubule) and/or nuclear skeleton also contribute to the biomechanical properties of cells. It is suggested to add some discussion about this.

Formats of editing and typesetting for figures and table in this manuscript are not well-organized. It needs to be improved.

Author Response

Q.1. In the results of AFM, it is suggested to provide the force-distance curves. The result showed that the dependencies of the elastic modulus on the indentation depth differ significantly for BT-20 and ZR-75 cells. Authors need to discuss the clinical significance in detail of these different biomechanical profiles.

It is well documented in the literature that cancer cells' mechanical properties differ from those of the healthy ones. Most of the measurements in this research field refer to cancer cell's average stiffness (average Young's modulus) and its comparison with the stiffness of their healthy counterparts. In this contribution, we demonstrate that the cellular stiffness varies as a function of the depth at which it is measured, and the character of its depth dependence changes with a cancer cell type. We, therefore, have a new parameter to define cancer cell mechanical properties that is more precise than an average stiffness value. The actin cytoskeleton is the main cellular component that determines the cell's mechanical properties, displacements, and dynamics. The difference in the organization of the actin cytoskeleton and the spatial distribution of mechanical properties between the cells of different types of breast cancer indicates a difference in the mobility of cells and their ability to contact each other. We suspect that cancer cells with different invasive potentials could have different actin cytoskeletons, and their mechanical properties should differ. With this novel approach, we have a tool to explore a putative correlation between invasive potential and the nanomechanical properties of cancer cells. This approach may support the development of novel diagnostic/predictive tools in the frame of clinical cancerology. Moreover, the high sensitivity of the actin cytoskeleton of BT-20 cancer cells to cytochalasin D treatment may be a basis for the development of therapy of this molecular cell-biological sub-type of breast cancer based on actin-cytoskeleton-targeting drugs for inhibiting the cell motility.

Q.2. In this study, the authors focused in role of actin cytoskeleton in the mechanical properties of two molecular genetic types of breast cancer cells. There is a possibility that other cytoskeleton (e.g. microtubule) and/or nuclear skeleton also contribute to the biomechanical properties of cells. It is suggested to add some discussion about this.

We inserted the following fragment into the manuscript: “In this work, we focused on the spatial actin cytoskeleton structures in BT-20 and ZR-75 cancer cell lines because their difference may explain different characters of the dependence of Young’s modulus on the indentation depth for the cells. The actin cytoskeleton makes a major contribution to cell biomechanical parameters measured by ACM [58]. The effect of microtubule-targeting drugs is more difficult to detect by AFM. Due to their location in the depth of the cells, they play a less important role in the cellular mechanical properties as measured by AFM [58, 59]. Two cancer cell lines differ in an intermediate cytoskeleton: ZR-75 cells have a vimentin cytoskeleton, and BT-20 cells have a cytokeratin cytoskeleton. However, the intermediate cytoskeleton spatial structure does not explain clearly a sharp change in the ratio of Young’s moduli of two cell lines within the 200-300 nm cell surface layer.” New references were added to the Reference list.

Q.3. Formats of editing and typesetting for figures and table in this manuscript are not well-organized. It needs to be improved.

In order to improve the reader's perception of Figures, we have carefully reviewed them and made the following changes: Figure 2 - We represented the AFM images without Color bar and Data scale (the scan size is present in Figure 2 capture). Figure 3 - Cell lines are marked with different colors. Figure 4 - Figure was re-organized, words “Cell line, Fixed and dried cells, Live cells” were marked with different colors.